# BECOME A PROFICIENT PLAYER WITH LIMITED DATA THROUGH WATCHING PURE VIDEOS

**Weirui Ye**[123*]  **Yunsheng Zhang**[23*]  **Pieter Abbeel**[4]  **Yang Gao**[123†]

Tsinghua University[1], Shanghai Artificial Intelligence Laboratory[2]
Shanghai Qi Zhi Institute[3], UC Berkeley[4]
`ywr20@mails.tsinghua.edu.cn, ys-zhang18@tsinghua.org.cn`
`pabbeel@berkeley.edu, gaoyangiiis@tsinghua.edu.cn`

## ABSTRACT

Recently, RL has shown its strong ability for visually complex tasks. However, it suffers from the low sample efficiency and poor generalization ability, which prevent RL from being useful in real-world scenarios. Inspired by the huge success of unsupervised pre-training methods on language and vision domains, we propose to improve the sample efficiency via a novel pre-training method for model-based RL. Instead of using pre-recorded agent trajectories that come with their own actions, we consider the setting where the pre-training data are action-free videos, which are more common and available in the real world. We introduce a two-phase training pipeline as follows: for the pre-training phase, we implicitly extract the hidden action embedding from videos and pre-train the visual representation and the environment dynamics network through a novel forward-inverse cycle consistency (FICC) objective based on vector quantization; for down-stream tasks, we finetune with small amount of task data based on the learned models. Our framework can significantly improve the sample efficiency on Atari Games with data of only one hour of game playing. We achieve 118.4% mean human performance and 36.0% median performance with only 50k environment steps, which is 85.6% and 65.1% better than the scratch EfficientZero model. We believe such pre-training approach can provide an option for solving real-world RL problems. The code is available at `https://github.com/YeWR/FICC.git`.

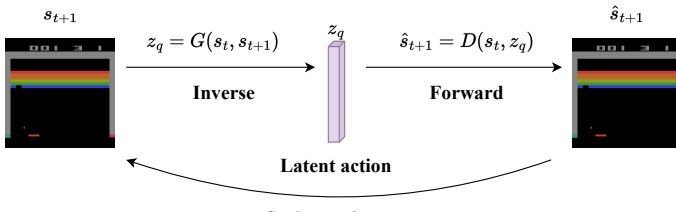

Figure 1: The forward-inverse cycle consistency builds an unsupervised training objective for model-based reinforcement learning from pure videos.

## 1 INTRODUCTION

Recently, deep reinforcement learning algorithms have achieved great success on various tasks, including simulated games, robotics manipulations, protein structure analysis and even controlling nuclear fusion (Schrittwieser et al., 2020; Jumper et al., 2021; Degrave et al., 2022). However, the great success of these RL algorithms is based on huge amounts of data. For example, it requires over 20 million games on Go for AlphaZero (Silver et al., 2017), and Liu et al. (2021) spend millions of data for playing simulated humanoid football. But in real applications or complex tasks, it is impossible to acquire such amounts of data through interactions with environments.

To keep strong performance while requiring much less data, some researchers propose to use model-based reinforcement learning (MBRL) algorithms. They build environmental world models in assis-

---

*Equal contribution
†Corresponding author

tance of planning to increase the sample efficiency. And experiments have proved the high sample efficiency of MBRL (Kaiser et al., 2019; Hafner et al., 2019). The high sample efficiency of MBRL shows the great potential for handling sequential decision-making problems in complex simulated environments and real-world (Hafner et al., 2020; Ye et al., 2021).

Although MBRL has improved sample efficiency a lot, it still requires a non-trivial amount of interactions to finish each task (Moerland et al., 2020; Schrittwieser et al., 2020). And each time, it learns from scratch, which makes it difficult to deploy quickly on different downstream tasks. Consequently, a good approach to improving this is to pre-train the world model with some data first. However, the datasets equipped with actions are hard to obtain on a large scale. This is because to collect a large-scale, high-quality dataset with actions, we need a good policy in the first place, and this becomes a chicken-and-egg problem. Instead, pure videos without action labels are more accessible and affordable in the real world. There are a huge amount of video datasets without action labels on the Internet.Thus, in this paper, we study how to pre-train the world models with action-free videos for MBRL.

We propose to pre-train a latent dynamics model based on inverse latent action prediction from pure videos without any action labels. We propose a novel cycle consistency loss by chaining the forward dynamics and the inverse dynamics, as shown in Figure 1. This loss can pre-train the forward models as well as the inverse models in the visual MBRL algorithms. Afterward, we fine-tune the downstream tasks based on the pre-trained models. Experiments show that our method can build sound representation and dynamics pretrained models for the downstream task. We achieve 118.4% mean human performance and 36.0% median performance with only 50k environment steps, which is 85.6% and 65.1% better than the scratch EfficientZero model. Our contributions are the following:

- We systematically study the problem of pre-training from action-free videos for model-based RL, which could be the foundation for future sample efficient, robust, and few-shot generalizable robots in the physical world.
- We propose a forward dynamics - inverse dynamics cycle consistency pre-training method that can jointly infer the latent actions from the video and train the representation function as well as the dynamics function. We also propose a practical fine-tuning scheme that achieves high performance on many downstream tasks.
- Our framework achieves the SoTA on the 60-minute Atari games and significantly outperforms others. Experiments show that the model pre-trained on distinct environmental data together can be fine-tuned well to the corresponding environments without re-pre-training.

## 2 RELATED WORKS

**Unsupervised Pre-training in NLP and CV** In recent work, researchers have found that the language model pre-trained with unsupervised learning can quickly and well generalized to down-steam language tasks (Devlin et al., 2018; Yang et al., 2019). Some researches show that the pre-trained model can be a good few-shot learner (Brown et al., 2020) or multi-task learner (Radford et al., 2019). More importantly, the two-stage procedure of training has become more popular for large models in NLP, such as Transformers (Vaswani et al., 2017; Brown et al., 2020). People find that in computer vision, similar unsupervised pre-training methods can build sound representation models for various visual tasks based on transformer (Li et al., 2019; Dosovitskiy et al., 2020). Contrastive learning and reconstruction are two common techniques to achieve these goals (He et al., 2020; Grill et al., 2020; He et al., 2022). Generally, all these methods aim to model a universal representation function, which can be fine-tuned well to some specific vision tasks, e.g. classification or detection.

**Unsupervised for Representation Learning in RL** Inspired by the great success of unsupervised pre-training in NLP and CV, researchers attempt to learn representations for visual RL in an unsupervised manner. People find that contrastive learning on online visual RL helps to extract good latent states for robotics control tasks (Laskin et al., 2020; Schwarzer et al., 2020). Ye et al. (2021) propose to improve the sample efficiency of model-based RL through temporal contrastive learning. Furthermore, Stooke et al. (2021) introduce a new unsupervised learning task to decouple the representation learning from policy learning. Xiao et al. (2022) propose to do better motor control from the masked visual pre-training method from real-world images. Besides, some researchers attempt to do pre-training and fine-tuning for RL down-stream tasks. Parisi et al. (2022) find that

ImageNet pre-trained visual representations can be competitive to ground-truth state representations for control policies. Schwarzer et al. (2021) propose to pre-train from offline datasets based on several unsupervised learning objectives and then fine-tune for down-stream tasks. For model-based algorithms, Deng et al. (2022) propose to pre-train the world model with unsupervised losses. All of the work above either study the model-free pre-training, or assume there are actions available during model-based pre-training. Our work studies the action-free model-based pre-training problem.

**Unsupervised learning from action-free videos** Pure videos or action-free videos are much more accessible and affordable in the real-world. Unsupervised learning from action-free videos has been studied in computer vision and imitation learning before. Dwibedi et al. (2019) and Wang et al. (2019) propose to utilize the temporal cycle consistency as the unsupervised learning objective for multiple vision tasks. Menapace et al. (2021) propose to predict action labels for playable video generation from unlabeled videos with image reconstruction objectives. In imitation learning, Edwards et al. (2019) studies the imitation from observation alone problem and proposes a similar unsupervised objective to ours. However, they do not work on the reinforcement learning problem. The closest work to ours is Seo et al. (2022), where they pre-train the model with evidence lower bound based on Dreamer-v2 (Yarats et al., 2021). However, unlike our approach, they do not explicitly infer latent actions.

## 3 PROBLEM FORMULATION

Before introducing our algorithm, we give some basic notations for the model-based reinforcement learning (MBRL) algorithms following conventions in Moerland et al. (2020). Firstly, the formal definition of Markov Decision Process (MDP) (Puterman, 2014) is defined as the tuple $(\mathcal{S}, \mathcal{A}, p, r, \gamma)$. Here, $\mathcal{S}$ is the state space, $\mathcal{A}$ is the action space, $p(s_{t+1}|s_t, a_t)$ is the probability function of the transition dynamics, $r_t = r(s_t, a_t)$ is the reward function corresponding to specific tasks, and $\gamma \in [0, 1]$ is the discount factor.

In model-based reinforcement learning algorithms, dynamics model learning is learned to approximate the transition dynamics of the environments. People aim to train a dynamics model $\mathcal{D} : \mathcal{S} \times \mathcal{A} \to \mathcal{S}$ through supervised learning. For the visual-based tasks, it is more computation-efficient to build a representation network $\mathcal{R} : \mathcal{O} \to \mathcal{S}$ to extract the latent states $\mathcal{S}$ from observations $\mathcal{O}$. Afterward, exploration techniques or planning algorithms can be applied based on the models $\mathcal{R}, \mathcal{D}$. One can utilize those models with model-free algorithms, such as predicting rewards.

In this work, we study the pre-training of $\mathcal{R}$ and $\mathcal{D}$ in model-based RL. Here $\mathcal{R}$ and $\mathcal{D}$ are task agnostic, and they are thus widely applicable to different downstream tasks. The $\mathcal{R}$ model can be learned from data, no matter with actions or without actions. However, $\mathcal{D}$ usually requires explicit actions as input, which is expensive to obtain in the real world. In this work, we study the task of pre-training $\mathcal{R}$ and $\mathcal{D}$ end-to-end from pure, i.e., action-free, videos.

## 4 METHOD

Compared to the data with action labels, pure videos are more common and accessible in the real world. However, pre-training the world models with such action-free data is challenging because we lack the ground truth action to train the action-based dynamics model. To address this dilemma, we propose a novel forward-inverse cycle consistency (**FICC**), an unsupervised training objective for pre-training the models in model-based RL. Our FICC framework is shown in Figure 2. It first employs an inverse dynamics model that infers latent action $z$ from the current state $s_t$ and next state $s_{t+1}$ from pure videos. With the inferred latent action $z$ and the current state $s_t$, a forward dynamics model maps them to the predicted next state $\hat{s}_{t+1}$. Our proposed forward-inverse cycle consistency loss is pulling the $\hat{s}_{t+1}$ close to $s_{t+1}$. Here the forward and inverse in FICC refer to forward dynamics and inverse dynamics, respectively. The cycle here refers to the $s_{t+1}$, where it is first mapped to the latent action $z$ and then mapped back to itself again. We will discuss the FICC framework in detail in Section 4.1. To avoid shortcuts, we propose a special design for the inverse model, called Latent Action Generator (LAG), which is discussed in Section 4.2. Finally, we will introduce how to use such a pre-trained model for the downstream tasks in Section 4.3.

## 4.1 Forward-Inverse Cycle Consistency in Pretraining

Figure 2 shows our overall forward-inverse cycle consistency framework. First, we use a convolutional neural network $\mathcal{R}$ to extract states from the observations. This allows us to have a compact and semantic meaningful latent state to operate on. The representation model also alleviates the computation cost for the inverse and forward dynamics models. Then an inverse dynamics model takes in $s_t$ and $s_{t+1}$ and output the inferred latent action $z$. Note that we do not require the inferred latent action $z$ to have a one-to-one correspondence to the ground truth action. The mapping from the actual action to latent actions will be later learned when fine-tuning on the downstream task. Finally, the forward model takes in the inferred latent action $z$ and the current state $s_t$ to predict the next state $\hat{s}_{t+1}$. We use a cosine similarity loss to pull the $\hat{s}_{t+1}$ close to the true state $s_{t+1}$.

Besides the cosine similarity cycle consistency loss, we find that adding extra reconstruction loss further helps to stabilize the training. We add two reconstruction losses. First, we reconstruct $o_t$ from $s_t$, which will ensure that all information has been encoded in the latent state. Second, we also reconstruct $o_{t+1} - o_t$ from $s_t$ and $z$. This helps the inverse dynamics model to focus on the changes in the environment. We also unrolled those losses for five steps, since the dynamics models usually unroll for multiple steps in the downstream tasks. In summary, the unsupervised pre-training loss is:

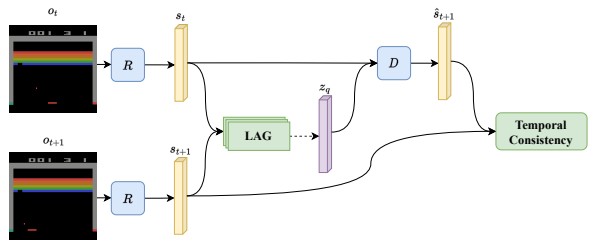

**Forward-Inverse Cycle Consistency (FICC)**

Figure 2: The temporal forward-inverse cycle consistency (**FICC**) for pre-training environmental models.

$$\mathcal{L}_{\text{cc}} = \overbrace{-\cos(\hat{s}_{t+1}, s_{t+1})}^{\text{cycle consistency}} - \underbrace{\ln p(o_{t+1} - o_t | s_t, z_q)}_{\text{difference reconstruction}} - \underbrace{\ln p(o_t | s_t)}_{\text{reconstruction}}, \tag{1}$$

where $s_t = R(o_t), s_{t+1} = R(o_{t+1}), z_q = \text{inverse}(s_t, s_{t+1}), \hat{s}_{t+1} = \mathcal{D}(s_t, z_q)$. The hyperparameters and more implementation details such as unrolling are in Appendix D.

## 4.2 Latent Action Generator based on Vector Quantization

There is one caveat in the FICC framework we introduced in Section 4.1, that is, there might be a shortcut in the cycle consistency. Since the latent action $z$ is a continuous vector, it can encode unlimited information theoretically if the scalar precision of z is infinite. Consequently, there is one shortcut that the latent action $z$ can encode most information of $s_{t+1}$ rather than the difference of $s_t$ and $s_{t+1}$. And the inverse model can directly copy all the information in $s_{t+1}$. Correspondingly, the forward model can learn an identity function. This shortcut will achieve zero loss in the cycle consistency we defined, but it won't be useful for the downstream task at all. The reason that the above shortcut happens is because the latent action $z$ contains too much information. We know that in practice, the action we take usually has a small amount of information. For example, in video games, the actions are some key strokes. In the robotics domain, the action can be represented by a few numbers describing the joint location and speed, etc.

In order to avoid this shortcut, we propose an inverse-dynamics model called Latent Action Generator (LAG) that regularizes the latent action $z$ to have a limited amount of information. People have proposed several techniques to enforce information bottlenecks in the neural networks, such as using low dimensional features, variational information bottlenecks (Alemi et al., 2016), or vector quantizations (Van Den Oord et al., 2017). In theory, we can use any of those techniques here to avoid the shortcut. In this paper, we explore using vector quantizations for this purpose in favor of their scalability and ease of training (Van Den Oord et al., 2017).

Figure 3 shows our proposed LAG model. First, a CNN takes in $s_t$ and $s_{t+1}$, outputting a latent action embedding $z_e \in \mathcal{R}^D$. We also define a dictionary of vector quantization embeddings $e = \{e_1, e_2, \cdots, e_k\}$. The dimensionality of each vector in the dictionary is the same as $z_e$. The output of LAG is defined as the nearest dictionary embedding to $z_e$, denoted as $z_q$. The $z_q$ is the latent action

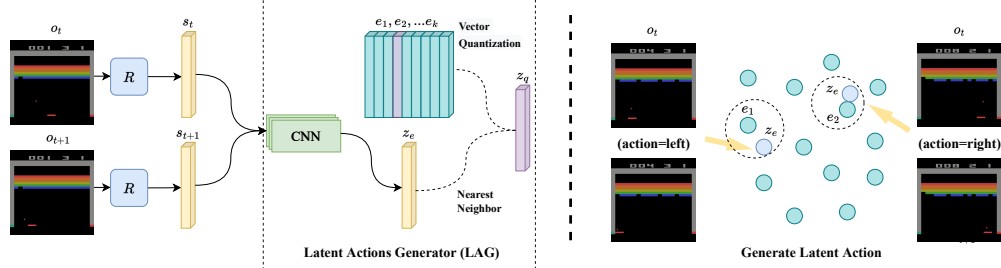

Figure 3: Left: Acquire the latent action embeddings through vector quantization based on the adjacent frames. Right: Illustration of the latent action representations. Different actions will be mapped into different embeddings.

we discussed in Section 4.1. In the standard VQ-VAE, there are multiple quantized embeddings for a single image. However, we find that using only one embedding is enough in our case.

The training of LAG is similar to VQVAE (Van Den Oord et al., 2017), where the codebook loss brings the code closer to the encoder output, and the commitment loss prevents the encoder from switching between different codes. The training objective of the LAG is:

$$\mathcal{L}_{\text{vq}} = ||\text{sg}[z_e(s_t, s_{t+1})] - e||_2^2 + \beta\, ||z_e(s_t, s_{t+1}) - \text{sg}[e]||_2^2 \tag{2}$$

where $z_e$ denotes the CNN that outputs $z_e$, sg denotes the stop-gradient, $\beta$ is the commitment trade-off, and it is set to 1. We conduct the ablation study to show the significance of the information bottleneck (Section 5.4). More implementation details are attached in Appendix D.

Finally, the unsupervised training objective in the pre-training phase is $\mathcal{L} = \mathcal{L}_{\text{cc}} + \alpha\mathcal{L}_{\text{vq}}, \alpha = 1$.

## 4.3 FINE-TUNING WITH ACTION ADAPTER

During fine-tuning on down-stream tasks, the agent interacts with the environment with real actions. However, the dynamics model is pre-trained via the latent action embedding. The only gap between them is the format of actions. Fine-tuning can be quite straightforward if each real action $a_t$ is mapped to the corresponding latent action embedding $e_k$ after the representation model $\mathcal{R}$ and dynamics model $\mathcal{D}$ are pre-trained. For convenience, we name such map $\mathcal{M} : \mathcal{A} \to R^D$ as action adapter. Then the dynamics model can predict the next states with the action adapter: $\hat{s}_{t+1} = \mathcal{D}(s_t, \mathcal{M}(a_t))$.

We statistically build such adapter during fine-tuning as Algo. 1 shows.

**Algorithm 1** Building Action Adapter

1: Given pre-trained LAG, embedding codebook $e$
2: $C[a, k] \leftarrow 0, a \in \mathcal{A}, k \in K; \mathcal{M} \leftarrow \text{Dict}\{\}$
3: **for** $t \in N$ **do**
4:     Obtain triple $(s_t, a_t, s_{t+1})$
5:     $e_k = \text{LAG}(s_t, s_{t+1})$
6:     $C[a, k] \leftarrow C[a, k] + 1$
7: **end for**
8: Sort $T_{a,k} = (C[a, k], a, k), a \in \mathcal{A}, k \in K$ by $C[a, k]$
9: Such that $a < b, T_{a,k} \geq T_{b,k}$ and $k < l, T_{a,k} \geq T_{a,l}$
10: **for** $C[a, k], a, k \in T$ **do**
11:     **if** $a \in \mathcal{M}.\text{keys}() \vee e_k \in \mathcal{M}.\text{values}()$ **then**
12:         Continue
13:     **end if**
14:     $\mathcal{M}[a] \leftarrow e_k$
15: **end for**
16: **Return** $\mathcal{M}$

During fine-tuning, the embedding codebook $e$ is frozen for a stable inference of the dynamics model. When a state-action-state transition triple $(s_t, a_t, s_{t+1})$ is collected in self-play, the pre-trained LAG will infer the corresponding latent action $e_t$ and store the relationship tuple $(a_t, e_t)$. We map each real action to the most common latent action. We also require different real actions to map to different latent actions. Please see Appendix D.3 for more implementation details.

With the pre-trained $\mathcal{R}$, $\mathcal{D}$ and the action adapter $\mathcal{M}$, we fine-tune them on the downstream tasks. In this work, we use EfficientZero (Ye et al., 2021) as the model-based RL algorithm because of its high performance with limited data. EfficientZero is composed of the representation network $\mathcal{R}$, the dynamics network $\mathcal{D}$, the policy network $\pi$, the value network $V(s)$ and the reward predictor $r(s, a)$. Given the pretrained $\mathcal{R}$ and $\mathcal{D}$, we fine-tune with the two learning rates: the small fine-tune learning rate $l_f$ for the pretrained components, and the large learning rate $l_s$ for the components that are not initialized ($\pi, V(s), r(s, a)$).

## 5 Experiments

As mentioned above, we propose three key components for handling the pre-training and fine-tuning with pure videos: 1) forward-inverse cycle consistency; 2) latent action generator; 3) statistical action adapter for mapping the real actions to latent actions. In this section, we aim to answer the following two questions: 1) Whether the pre-training can help downstream RL tasks; 2) What is the importance of each component. We conduct systematic experiments to answer those questions.

### 5.1 Experimental Setup

**Environments** Recently, sample efficiency has become an important topic in RL. Kaiser et al. (2019) introduce the Atari 100k benchmark, including 26 games. This benchmark consists of 100k steps of interactions with environments, which is equivalent to two hours of human game playing. This benchmark is a popular benchmark for evaluating the sample efficiency of RL algorithms due to the complex visual observations and diverse tasks (Kaiser et al., 2019; Kielak, 2020; Srinivas et al., 2020; Kostrikov et al., 2020; Ye et al., 2021). Recent developments in sample efficient RL (Ye et al., 2021) have achieved super-human performance on the Atari 100k benchmark. In order to further challenge the RL algorithm, we propose to use the Atari 50k benchmark, which only consists of 50k steps or one-hour game-play of interactions. We conduct the experiments on top of EfficientZero, which is the current SoTA model-based algorithm on the Atari 100k benchmark.

**Hyper-parameters** The hyper-parameters for pre-training are listed in Appendix B. For fine-tuning, we train for 50k steps and follow other hyper-parameter settings in EfficientZero, which are listed in the Appendix C. We update the action adapter every 1000 transitions.

**Evaluation** Mnih et al. (2015) provide a baseline of human sample-efficiency, and we follow the common evaluation metric for the 26 Atari games, namely human-normalized score (HNS). It is defined as $(\mathrm{score_{agent}} - \mathrm{score_{random}})/(\mathrm{score_{human}} - \mathrm{score_{random}})$. In each game, we average the scores over 100 evaluations across 3 runs with different seeds. Following the previous works, we report the median (Mdn) and mean (Mn) HNS over the 26 games. And we use the statistical tools proposed by Agarwal et al. (2021) to quantify uncertainty.

**Pretraining Dataset** We use the EfficientZero replay buffer as the pre-training dataset. We train the EfficientZero for 1M transitions from scratch and save the replay buffer as the pre-training dataset.

**Baselines** We compare the following baselines. EfficientZero (Ye et al., 2021) trained from scratch: since we use EfficientZero as our base model-based RL method, we train from scratch for the 50k environment steps as the baseline. To the best of our knowledge, there are few works studying pre-training with action-free videos on Atari games. SGI (Schwarzer et al., 2021) digs into the similar learning paradigm and achieves high sample-efficiency on Atari games. We also compare to SGI, which proposes pre-training with action-labeled data on Atari games. All the comparisons are under the same setting of 50k environment steps. We note that SGI is an "oracle" method since it uses ground truth actions during pre-training.

### 5.2 Performance Comparison

**HNS Scores** We compare our method with the above baselines on 26 Atari games for 50k environment steps. As illustrated in Table 1, our method outperforms the previous SoTA method EfficientZero(EZ) on a large scale in such limited data settings. **20** of total 26 games get superior performance with our pre-trained models. Specifically, we achieve 0.360 median HNS as well as 1.184 mean HNS and outperform the human level in 6 games. Compared to SGI, a method with access to ground truth actions during pre-training, we also outperform it by a large margin. Agarwal et al. (2021) propose to use the inter-quartile mean (IQM) normalized score and quantify the uncer-

Table 1: Comparisons of HNS on 26 Atari 50k for fine-tuning with 3 different runs.

| Method | Median | Mean | IQM | require action |
|---|---|---|---|---|
| EZ scratch | 0.218 | 0.638 | 0.208 | w.o. pre-training |
| SGI | 0.132 | 0.739 | 0.290 | ✓ |
| EZ + FICC (ours) | **0.360** | **1.184** | **0.353** | ✗ |

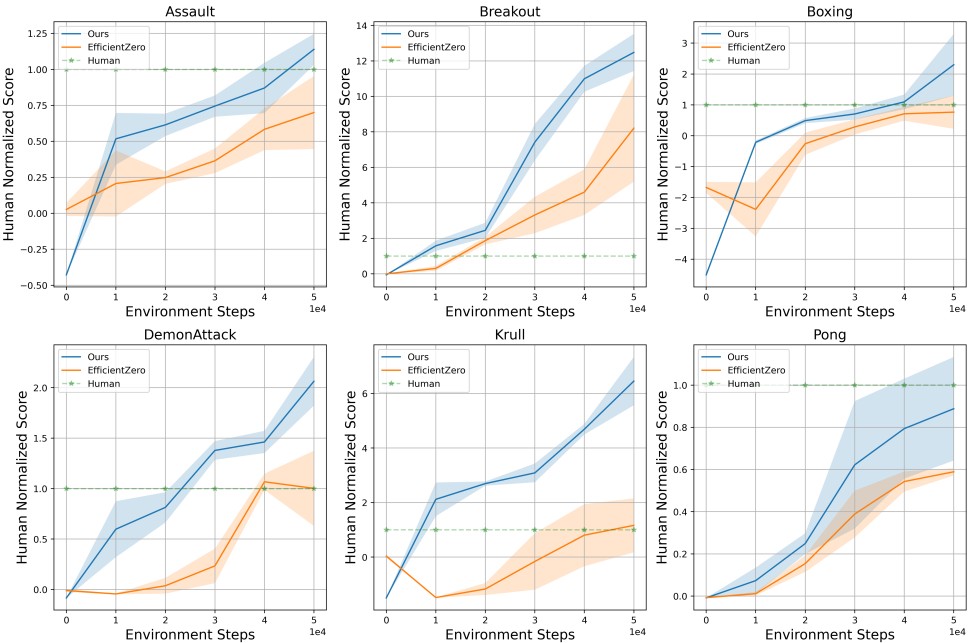

Figure 4: Learning curves on 6 out of 26 Atari games. We compare the curves of vanilla EZ and EZ with our pre-trained model from action-free videos. It indicates that our pre-trained model is able to improve the sample efficiency during the fine-tuning process.

tainty via percentile bootstrap confidence intervals. And our method is much superior on Atari 50k, including all these metrics.

**Sample Efficiency** Figure 4 shows the learning curves of our method and the baseline EZ method on six Atari games. Here the orange curve is the vanilla EZ, while the blue curve is EZ with our pre-training method. We find that our pre-training method consistently improves the HNS scores in all environments. Further, our method outperforms the baseline under almost any amount of data.

## 5.3 SHARING PRETRAINING AMONG MULTIPLE ENVIRONMENTS

The pre-training model we experimented with in Section 5.2 are trained per environment. That is, each pre-trained model only sees the data from the downstream task. However, in natural language processing and the image recognition domain, large-scale pre-trained models are only trained once and later applied to many different downstream tasks. A natural question is that, can our framework supports training one model and fine-tune on multiple tasks?

In this part, we aim to answer this question. We select a set of 6 environments (as shown in Table 2), and pre-train a single model with our method on all data from them. To accommodate the larger scale of pre-training data size, we use 3 residual blocks instead of 1 in the representation and dynamics model. All other settings are the same as before. We name this model as EZ-L, where L denotes "large". Firstly, we pre-train the EZ-L model across all the 6 environmental datasets and obtain the action embedding codebook $e$. Then we do fine-tuning on the 6 environments respectively based on the pre-trained EZ-L model. For each task, we build the corresponding action adapter $\mathcal{M}$ during fine-tuning. In this way, we pre-train one model and fine-tune it on multiple tasks. Here we follow

Table 2: Pretrain 6 games with different styles and rules in a single model, and then fine-tune each game with the same pre-trained model (EZ-L). Experiments show that EZ-L achieves superior performance compared to EZ scratch on most games.

| Games | Boxing | Breakout | CrazyClimber | MsPacman | Pong | Qbert |
|---|---|---|---|---|---|---|
| EZ scratch | 0.597 | 8.211 | 1.183 | 0.040 | 0.589 | 0.199 |
| EZ with FICC | **1.949** | **12.477** | **1.533** | **0.051** | 0.888 | **0.305** |
| EZ-L with FCC | 1.230 | 9.361 | 1.133 | 0.051 | **1.099** | 0.279 |

exactly the same fine-tuning hyper-parameters. The results are listed in Table 2. We found that EZ-L improves upon EZ scratch baseline in 5 out of 6 environments. In Pong, it even outperforms the environment-specific pre-training by a large margin. In summary, we show that it is possible to have only a single pretrained model and get the benefits of the pre-training in most environments.

## 5.4 ABLATION STUDY

In this section, we will conduct some ablation studies to investigate the two significant components of our method, the forward-inverse cycle consistency in Sec. 4.1 and the vector quantization for latent action generation in Sec. 4.2. We choose four different games for this ablation. We run each experiment for three different seeds with 100 evaluations. Other hyper-parameters are all the same. We make more ablation studies about the datasets and dimensions of latent action $z$ in App. A.

**FICC and Reconstruction Objectives Ablation** Here we do ablation studies to further investigate the effect of the FICC and the reconstruction objectives mentioned in Eq. 1. As for the FICC ablation, we remove the image difference reconstruction objective $\ln p(o_{t+1} - o_t | s_t, z_q)$. To show the individual performance delta of each reconstruction loss, we remove the image reconstruction objective $\ln p(o_t | s_t)$. The results are shown in Table 3. We find that the removal of the image difference cycle consistency loss has deteriorated the performance by a large margin. But it is still better than the EZ from scratch baseline. This supports our claim that this loss encourages the inverse dynamics to focus on the changes in the environment. Moreover, the model pre-trained without OR has comparable performance to the Pre-train with FICC. Therefore, the cycle-consistent difference reconstruction is more significant and it can work well without OR. Compared with FICC and FICC without OR, the former has more advantages in most environments, except that the latter has obvious performance improvement in the Boxing environment. We believe that some properties in the Boxing environment make the model learn harmful representations from observation reconstruction.

**VQ Avoids Shortcut Learning Ablation** In Sec. 4.2, we discuss the need for the information bottleneck. Here we ablate this component to investigate the significance of it. The latent action generator takes in the adjacent states and outputs one embedding to represent latent actions. We build a LAG to generate latent action without the quantization step. The output of the neural network $z_e = z(s_t, s_{t+1})$ is directly fed into the dynamics model instead of utilizing the vector quantization to replace $z_e$ with $z_q$. For convenience, we name such LAG with direct outputs as LAG-D and the LAG with VQ technique as LAG-VQ. Here "D" is short for direct. Following the previous ablation environments and parameters, the results are listed in Table 4. For some games, the LAG-D keeps comparable performance to LAG-VQ (Breakout). However, for some games, such as Krull underlined in the table, LAG-D performs even worse than the scratch model. It indicates that the model pre-trained without the information bottleneck harms the fine-tuning on some down-stream tasks, which can be caused by short-cut learning. We notice that in some game, such as Pong, the LAG-D is slightly better than the LAG-VQ. However, from the perspective of most games, LAG-VQ is much better than LAG-D.

Table 3: Ablation of the cycle consistency objectives for 50k environments steps fune-tuning. The cycle consistency improves the performance of all four evaluation games.

| Games | Boxing | Breakout | Krull | Pong |
|---|---|---|---|---|
| EZ scratch | 0.597 | 8.211 | 1.164 | 0.589 |
| Pre-train with FICC | 1.949 | **12.477** | **6.455** | **0.888** |
| FICC without Observation Reconstruction (OR) | **6.002** | 12.128 | 3.692 | 0.816 |
| FICC without Difference Reconstruction (DR) | 1.160 | 11.156 | 3.367 | 0.648 |

Table 4: Ablation of the types of LAG for 50k environment steps fune-tuning. LAG-VQ is the original implementation, while LAG-D outputs the latent action directly from the neural networks. For games like Krull, LAG-D behaves even worse than the scratch model. LAG-D fails to build an information bottleneck, which might cause some short-cut learning during pre-training.

| Games | Boxing | Breakout | Krull | Pong |
|---|---|---|---|---|
| EZ scratch | 0.597 | 8.211 | 1.164 | 0.589 |
| LAG-VQ | **1.949** | **12.477** | **6.455** | 0.888 |
| LAG-D | 1.174 | 12.435 | 0.712 | **1.095** |

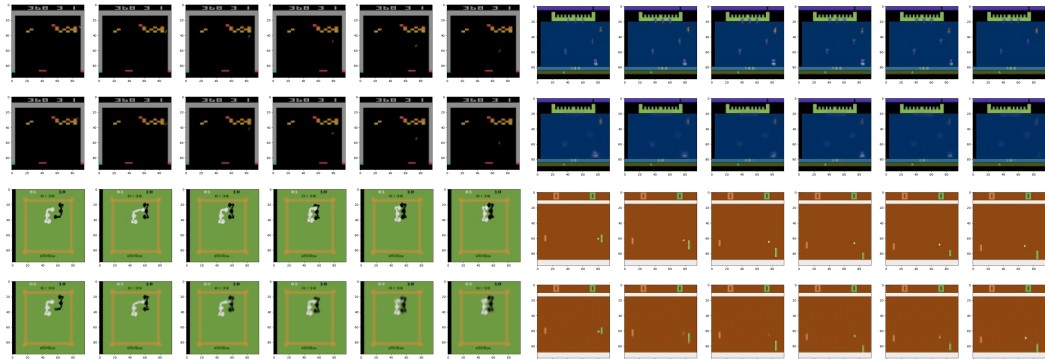

Figure 5: Dynamics prediction visualization of the pre-trained model. Top left: Breakout. Top right: Krull. Bottom left: Boxing. Bottom right: Pong. The first line of each environment is the ground truth images, and the second line is the images predicted and generated by the model.

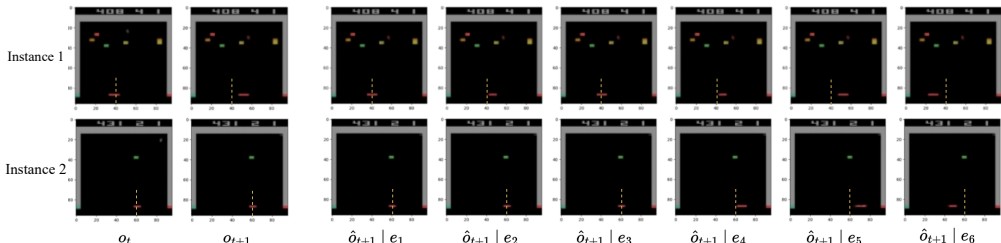

Figure 6: Visualization of the latent action effects. Each row in the figure shows a different observation. The yellow dashed line is the anchor of the board position at time step $t$. The left two images of each line are the current observation $o_t$ and the ground truth next step observation $o_{t+1}$. The six figures on the right are the reconstructed images with the index 1 to index 6 latent actions as the input to the dynamic function, which results in distinct predictions $\hat{o}_{t+1}$.

## 5.5 VISUALIZATION ANALYSIS

In this section, we do some visualizations to verify the effectiveness of the latent action generator (LAG) as well as the dynamics model based on the latent actions.

**Dynamics Prediction** In Figure 5 we visualize the dynamic function output of the pre-trained model with 5 unroll steps. In each step $t$, we use the LAG to get latent action embedding $z_t$ from the current state $s_t$ and next state $s_{t+1}$, and use a decoder to reconstruct the output. The whole process is consistent with the pre-training algorithm. From the results, we observe that the pre-trained model can make high-quality reconstruction for the images of the next several steps. The prediction ability of the model includes the motion of objects in the scene (the ball in Breakout and Pong), the motion of the agent itself (Breakout's guard, Pong's right guard, Boxing's left character), and the motion of other characters (Boxing's right character). This shows that our LAG can effectively encode the differences between the two states.

**Latent Action Embeddings** The effects of latent actions are visualized in Figure 6. Here we select the game Breakout and visualize 6 of the total 20 latent actions. Results show that the latent actions have the same meanings for different rows. For example, latent action with index 6 indicates that the guard moves to the left, while actions with index 5 indicate that the guard moves to the right.

## 6 DISCUSSION

In this paper, we propose to pre-train the environment models via a novel forward-inverse cycle consistency from the action-free videos. To achieve this, a vector quantization based latent action generator is introduced to avoid the shortcut learning. We evaluate the pre-trained model via fine-tuning for Atari games with limited 50k environment steps. Experiments prove the efficiency of our method. As for the limitations, current architecture is difficult to handle the continuous action space. We leave this as future work.

# 7 REPRODUCIBILITY STATEMENT

The main implementations of our proposed method are in Section 4.1, 4.2 and 4.3. In addition, the settings of the experiments and hyper-parameters we choose are in Appendix B. And the implementation details are in Appendix D.

# 8 ACKNOWLEDGMENTS AND DISCLOSURE OF FUNDING

This work is supported by the Ministry of Science and Technology of the People´s Republic of China, the 2030 Innovation Megaprojects "Program on New Generation Artificial Intelligence" (Grant No. 2021AAA0150000). This work is also supported by a grant from the Guoqiang Institute, Tsinghua University.

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

# A  MORE ABLATION STUDIES, COMPARISONS, AND VISUALIZATIONS

## A.1  ABLATION: DIFFERENT DATASETS

Currently, the dataset used in this work is from the EfficientZero replay buffer. It is important to investigate the impact of pre-training data with different collecting methods. To measure this, we make ablations for different pre-training datasets under the same settings. We compare the performance of the models pre-trained from EZ replay buffer, random trajectories, DQN weak dataset, and expert trajectories. As for DQN weak dataset, which is similar to SGI, we choose the first 1M steps from the DQN Replay dataset. As for the expert trajectories, we rollout an expert agent repetitively with 25% random actions to increase dataset diversity. Here, each dataset contains 1M transitions. We keep the same experimental setting and choose the same 4 environments with three different seeds for fine-tuning on Atari 50k. The results are as Table 5 shows:

Table 5: Ablation of different collected datasets for 50k environment steps fine-tuning. There are four kinds of datasets: EZ Replay buffer, Random trajectories, DQN Weak dataset (first 1M transitions), and Expert trajectories. The model pre-trained from Random trajectories can perform well. And our method can fit with datasets collected by distinct agents.

| Games | Boxing | Breakout | Krull | Pong |
|---|---|---|---|---|
| EZ scratch (No pre-training) | 0.597 | 8.211 | 1.164 | 0.589 |
| EZ Replay buffer | 1.949 | **12.477** | **6.455** | 0.888 |
| Random trajectories | 2.216 | 11.604 | 4.136 | 0.736 |
| DQN Weak dataset | **3.279** | 10.422 | 5.566 | **0.942** |
| Expert trajectories | 2.785 | 11.655 | 5.159 | 0.768 |

Notably, SGI cannot work under the random dataset or the DQN weak dataset. But our method can work. Specifically, we have three conclusions: (1) Our pre-training method is not sensitive to the quality of datasets. Even if pre-trained under random trajectories, the model has improved significantly among the 4 environments. (2) Our pre-training method is not sensitive to the data-collection method. In terms of the DQN weak dataset, our method can still work well and even outperform the model pre-trained under EZ Replaybuffer in 2 environments. (3) Expert trajectories are not the best choice of pre-training dataset in our method.

## A.2  ABLATION: DIMENSIONS OF LATENT ACTION $z_e$ OF LAG-D

In Sec. 5.4, we compare the performance between LAG and LAG-D which generates the latent actions without vector quantization. And results show that the model pre-trained with LAG-D can cause shortcut learning and harm the fine-tuning in some games. Therefore, another question is: can the shortcut learning be avoided for LAG-D through simply reducing the dimension of the latent action $z_e$. To answer this, we reduce the dimension of $z_e$ from 5 to 2 and use the LAG-D for pre-training. We keep the same experimental setting and choose the same 4 environments with three random seeds. The results are listed in Table 6. After reducing the dimension of $z_e$, the LAG-D still fails in the game Krull. It indicates that simply reducing the dimension of $z_e$ cannot avoid shortcuts.

Table 6: Ablation of different dimension of $z_e$ in LAG-D for 50k environment steps fine-tuning. Here we reduce the dimension of $z_e$ to 2 and the LAG-D still fails in the game Krull.

| Games | Boxing | Breakout | Krull | Pong |
|---|---|---|---|---|
| EZ scratch | 0.597 | 8.211 | 1.164 | 0.589 |
| LAG-VQ (dim=5) | **1.949** | **12.477** | **6.455** | 0.888 |
| LAG-D (dim=5) | 1.174 | 12.435 | 0.712 | **1.095** |
| LAG-D (dim=2) | 1.465 | 10.925 | 0.759 | 0.741 |

## A.3  COMPARISONS OF MORE ALGORITHMS

Apart from the comparison to SGI, we also compare our method with APV Seo et al. (2022), which proposes to stack an action-free dynamics model before the original dynamics model based on

Dreamer V2 Hafner et al. (2020). We try to run their method on the Atari benchmark, however, we find that it completely fails under the 50k sample limit. In order to achieve non-zero performance, we use 10x data during fine-tuning, namely Atari 500k. As for pre-training data for APV, we choose the best choice suggested by Schwarzer et al. (2021): random sample 3M samples from checkpoints throughout the DQN training. All the other training details follow the original paper. We run on the same 4 ablation environments used in our paper and aggregate the results with three seeds. The results are shown in Table 7. In general, our method has superior performance than APV because we are able to infer the latent actions during pre-training. But APV pre-trains the model without action conditions.

Table 7: Comparison between our method and APV. Our method outperforms APV in general.

| Games | Boxing | Breakout | Krull | Pong |
|---|---|---|---|---|
| EZ scratch; 50k | 0.597 | 8.211 | 1.164 | 0.589 |
| EZ Pre-train with FICC; 50k | **1.949** | **12.477** | **6.455** | **0.888** |
| APV; 50k | 0.057 | -0.013 | 0.623 | 0.020 |
| ALV; 500k | 0.186 | **0.080** | **35.028** | 0.161 |

## B    HYPER-PARAMETERS FOR PRE-TRAINING

The hyper-parameters of pre-training are listed in Table 8. The 1 million pre-training data is collected from EfficientZero training from scratch.

| Parameter | Setting |
|---|---|
| Observation down-sampling | $96 \times 96$ |
| Frames stacked | 4 |
| Frames skip | 4 |
| Minibatch size | 256 |
| Optimizer | SGD |
| Optimizer: learning rate | 0.02 |
| Optimizer: momentum | 0.9 |
| Optimizer: weight decay | $10^{-4}$ |
| Learning rate schedule | $\cos 0.02 \to 0.0002$ |
| Max gradient norm | 10 |
| Training steps | 50K |
| Pre-training data | 1M |
| Unroll steps for training dynamics | 5 |
| Shape of the state | $64 \times 6 \times 6$ |
| VQ: Number of latent action embeddings | 20 |
| VQ: Dimension of latent action embeddings | 5 |

Table 8: Hyper-parameters for FICC pre-training.

## C    HYPER-PARAMETERS FOR FINE-TUNING

During fine-tuning, we follow the same training hyper-parameters of EfficientZero except for reducing the learning rate of the pre-trained models, namely the representation model and the dynamics model. For the pre-trained model, we choose 0.05 as the initial learning rate while we choose 0.2 for those un-pretrained models. We use SGD optimzier as well as constant learning rate scheduler, which is also the same as EfficientZero.

## D    IMPLEMENTATION DETAILS

### D.1    NETWORK DESIGN DETAILS

The design of the representation and dynamics is consistent with EfficientZero since we use it as the model-based algorithm on the downstream tasks. The implementation details can be founded at (Ye et al., 2021) Appendix A.1. The representation network takes a $3 \times 96 \times 96$ tensor as input, which is down-sampled from Atari game image, and outputs the state tensor with shape $64 \times 6 \times 6$. The dynamics network takes the current state $s_t$ and current action $a_t$ as input and output the next state $s_{t+1}$ with shape $64 \times 6 \times 6$. Regarding dynamics network, the only difference between us and EZ is that the shape of the input action tensor is $5 \times 6 \times 6$ instead of $1 \times 6 \times 6$ because we map the ground truth action to the latent action embedding.

The design of the LAG is listed here:

- Concatenate the current state $s_t$ and the next state $s_{t+1}$ into 128 planes.
- 1 convolution with 64 output planes, with BN and ReLU
- 1 residual block with 64 planes.
- 1 convolution with 5 output planes, with BN and ReLU
- Flatten the planes with shape $5 \times 6 \times 6$ into 180 dimensions vector.
- 1 fully connected layers with 5 output dimensions, with is the dimension of latent action embeddings.
- 1 VQ model to quantized the vector to an action embedding.
- Expand dimension at the end of the action embedding and repeat 6 times to $5 \times 6 \times 6$.

Note that the pre-trained LAG is used and only used for makeup counting table $C$ during finetuning.

The design of the image decoder is listed here:

- 3 transposed convolution with 64 planes, stride 2, with BN and ReLU
- 1 convolution with 3 output planes.
- 1 Sigmoid activation to output a RGB image.

### D.2    UNROLLING DETAILS IN TRAINING

The unroll and loss calculation in our pre-training process is as follows: (1) Randomly select a trajectory from the video pre-training data and generate image sequences with the length of $T + 1$, namely $o_0, o_1, ..., o_T$ chronologically. (2) Obtain the state tensor using the representation network: $s_t = R(o_t)$ for $t$ from 0 to $T$. (3) Generate all latent action vectors using the latent action generator (LAG): $z_t = LAG(s_t, s_{t+1})$ for $t$ from 0 to $T - 1$, while building the training objective $\mathcal{L}_{vq,t}$ of the LAG using eq.(2). For convenience, we set the first predicted state $\hat{s}_0$ to be $s_0$ i.e. $\hat{s}_0 = R(o_0) = s_0$. (4) Predict next $T$ time-step state tensor using dynamics network and latent action: $\hat{s}_{t+1} = D(\hat{s}_t, z_t)$ for $t$ from 0 to $T-1$. Note that compared with the MuZero-style method, our dynamics network uses latent action $z_t$ as input rather than ground truth action $a_t$ because of our action-free pre-train setting. (5) Use reconstruction head and difference reconstruction head to predict $o_t$ and $d_t = o_{t+1} - o_t$ from $\hat{s}_t$ for $t$ from 0 to $T - 1$. The two reconstruction heads both consist of four transposed 2d-convolution layers. (6) Finally, calculate $\mathcal{L}_{cc,t}$ for $t$ from 0 to $T - 1$ using Eq.(1) and sum up the losses: $\mathcal{L} = \mathcal{L}_{cc} + \alpha \mathcal{L}_{vq} = \sum_{t=0}^{T-1} \left( \mathcal{L}_{cc,t} + \alpha \mathcal{L}_{vq,t} \right), \alpha = 1$ for backpropagation.

### D.3    ADAPTER DETAILS

In our implementation, we maintain the counting table $C$ in the self-play workers. At the end of each MCTS search, after selecting the action $a_t$ and getting the next state $s_{t+1}$, we follow line 5 to line 7 in Algorithm 1, put $s_t$ and $s_{t+1}$ into the pre-trained LAG to get the index $k$ of the latent action embedding and update the counting table $C$.

Table 9: Scores achieved on the Atari 50k benchmark.

| Game | Random | Human | EfficientZero | SGI | Ours |
|---|---|---|---|---|---|
| Alien | 227.8 | 7127.7 | 456.0 | **839.1** | 625.7 |
| Amidar | 5.8 | 1719.5 | 48.1 | **150.9** | 88.6 |
| Assault | 222.4 | 742.0 | 586.2 | 723.5 | **814.4** |
| Asterix | 210.0 | 8503.3 | 2185.2 | 410.3 | **2603.7** |
| Bank Heist | 14.2 | 753.1 | 84.3 | **779.5** | 75.4 |
| BattleZone | 2360.0 | 37187.5 | 9442.2 | 4370.0 | **6220.0** |
| Boxing | 0.1 | 12.1 | 7.3 | **30.3** | 23.5 |
| Breakout | 1.7 | 30.5 | 238.2 | 43.3 | **361.0** |
| ChopperCmd | 811.0 | 7387.8 | 727.3 | **826.7** | 651.7 |
| Crazy Climber | 10780.5 | 35829.4 | 40416.8 | **53437.7** | 49182.7 |
| Demon Attack | 152.1 | 1971.0 | 1975.3 | 1533.1 | **3906.3** |
| Freeway | 0.0 | 29.6 | 7.1 | 0.0 | **14.3** |
| Frostbite | 65.2 | 4334.7 | 235.3 | **311.8** | 245.6 |
| Gopher | 257.6 | 2412.5 | 1226.6 | 457.9 | **1172.1** |
| Hero | 1027.0 | 30826.4 | 4676.3 | 3767.8 | **11073.0** |
| Jamesbond | 29.0 | 302.8 | 82.4 | **295.7** | 191.3 |
| Kangaroo | 52.0 | 3035.0 | 224.0 | 320.3 | **344.0** |
| Krull | 1598.0 | 2665.5 | 2841.0 | 5900.7 | **8489.0** |
| Kung Fu Master | 258.5 | 22736.3 | 13363.3 | **18804.7** | 12985.7 |
| Ms Pacman | 307.3 | 6951.6 | 575.0 | **1443.2** | 644.1 |
| Pong | -20.7 | 14.6 | 0.1 | -1.4 | **10.7** |
| Private Eye | 24.9 | 69571.3 | 96.7 | **100.0** | 66.7 |
| Qbert | 163.9 | 13455.0 | 2813.3 | 1093.8 | **4219.7** |
| Road Runner | 11.5 | 7845.6 | 3143.3 | **8824.6** | 3317.0 |
| Seaquest | 68.4 | 42054.7 | 301.5 | 654.1 | **880.8** |
| Up N Down | 533.4 | 11693.2 | 3128.9 | **27720.0** | 4808.9 |
| Normed Mean | 0.000 | 1.000 | 0.638 | 0.739 | **1.184** |
| Normed Median | 0.000 | 1.000 | 0.218 | 0.132 | **0.360** |

The self-play workers will regularly communicate with the train worker to transfer the counting table $C$ they maintained. The train worker will add up all the counting table, execute line 9 to line 17 in Algorithm 1, then distribute the adapter $\mathcal{M}$ to each self-play worker and reanalysis worker.

## D.4 FULL RESULTS

The full results of the Atari 26 games with 50k environment steps are listed as follows. Here the EfficientZero trains from scratch with 50k interactions. As for SGI, we choose the SGI-M, which consumes 3M mixed of 50M DQN replay buffer and then finetune the SGI for 50k steps. SGI requires actions as input. In contrast, our model consumes 1M replay buffer of EfficientZero training from scratch to pretrain the environment models. Then we finetune for 50k steps.

