# OpenReview forum: "Become a Proficient Player with Limited Data through Watching Pure Videos"
_ICLR.cc/2023/Conference — ICLR 2023 poster_

### Official Review · Reviewer_Krh3 · 2022-10-21

**Confidence:** 3
**Correctness:** 3
**Technical Novelty And Significance:** 4
**Empirical Novelty And Significance:** 3
**Recommendation:** 8

**Clarity, Quality, Novelty And Reproducibility:**

The paper is well written, clear and easy to follow. The method is simple and leads to good performance. The paper uses standard benchmarks and promises to release code upon acceptance.

**Details Of Ethics Concerns:**

No ethics concerns

**Strength And Weaknesses:**

## Strengths

* The paper tackles an important problem - sample efficiency of RL agent training. The approach is based on learning a representation through unsupervised pre-training that can then easily be adapted when training the agent. The topic of the paper should be of interest to a large audience.

* The experimental evaluation is thorough and shows excellent results, improving over SGI that uses action labels during pre-training, which are not as easy to obtain as simple replays.

* The method is simple and general and should be easily adaptable to other tasks and methods.


## Weaknesses

### W1 Pretraining Dataset
The main motivation of the paper is that for pre-training it is easier to collect unlabelled videos than state-action pairs. The evaluation does not fully show the full extend of this, as the data comes from training an agent on the environment. Pragmatically, at that point training a second agent is not useful, as the data collection already provides a trained agent. It could be helpful to analyze this dependency by varying the data-collection method. One could use human data, youtube videos or a random agent and measure the effect on the performance.

### W2 Generalization
From the paper it is not clear if the model is pre-trained for each environment individually, or if the pre-training happens across all environments and only the agent training is done separately. The second scenario is of course a much stronger showcase of the benefits of the approach.

### Minor:
Eq 2: when $\beta=1$, the stop-gradient operator should be superfluous as the loss then has the same gradient as simply $\Vert z_e - e \Vert^2_2$.


**Summary Of The Paper:**

The paper targets the low sample efficiency of current reinforcement learning approaches. It focuses on a central question: can we leverage pre-trained data without actions to achieve a good initialisation? The experiments on Atari show consistent improvements using the proposed pretraining strategy that is based on cycle consistency.

**Summary Of The Review:**

The paper addresses an important problem and proposes a simple method that achieves excellent results. There are some weaknesses in the evaluation, which are however overall outweighed by the strengths, as the evaluation still demonstrates the applicability of the method sufficiently.

---

### Official Review · Reviewer_wEqK · 2022-10-24

**Confidence:** 3
**Correctness:** 3
**Technical Novelty And Significance:** 2
**Empirical Novelty And Significance:** 2
**Recommendation:** 5

**Clarity, Quality, Novelty And Reproducibility:**

The paper is fairly well written, and would likely be able to be reproduced from the provided details. The approach is , which would further aid the reproducibility. However, the approach is not especially novel, it is largely a combination of existing ideas, such as cycle-consistency and VQVAEs, but applied to a new task of pretraining RL networks.

**Strength And Weaknesses:**

Efficient RL learning is an important task, and this approach shows benefit over random initialized training. The ablations show that some of the components, e.g., reconstruction loss, are useful. However, there are no comparisons to any previous works, which makes it hard to know how well or efficient this approach is compared to others. The approach is only evaluated on relatively simple videos of atari games, how would this approach work for more complex or real videos? How much of the approach relies on the pretraining data? For example, if it was purely random trajectories vs. top human player trajectories, how would that influence the pretraining? In Table 2, it seems that EZ-L is usually worse, which seems to contradict the writing.


**Summary Of The Paper:**

This paper proposes an approach to pretrain a network to learn features for RL from videos of the task without actions. The network uses a reconstruction loss and a cycle-consistency loss to learn this. The approach is evaluated on several atari game tasks.

**Summary Of The Review:**

The paper proposes a method to pretraing RL networks. The approach is straightforward, though not especially novel. The experiments show the benefit over random init, but are lacking comparisons to other works.

---

### Official Review · Reviewer_1ysS · 2022-10-29

**Confidence:** 2
**Clarity, Quality, Novelty And Reproducibility:** 1. Clarity and quality are high.
2. N…
**Correctness:** 3
**Technical Novelty And Significance:** 3
**Empirical Novelty And Significance:** 3
**Recommendation:** 6

**Strength And Weaknesses:**

Strengths
1. Novelty, contributions and experimental evidence are well-linked.
2. Paper is clear and concise.

Weaknesses
1. Novelty is quite limited in the sense that pre-training is well-established concept for other learning paradigm. However, to my knowledge, it was not applied to RL as the paper proposed.
2. It is not clear why paper only experimented on Atari 50k when Atari 100k is the standard benchmark. Given the efficiency of the proposed method, it should not be difficult to also extend the experiments to 100k. This will also allow direct comparisons for other SOTA methods.
3. Seo et al was cited as a comparable method. But no experimental was done to compare. Why?

**Summary Of The Paper:**

Paper proposes a framework which can use action-free videos (instead of action-tagged videos) to train RL models by a 2-phase pipeline. In phase 1, the approach infers the hidden action embedding from the videos, pre-train the visual representation and the predicted next frames. A vector-quantization step was also introduced to prevent the model from learning the shortcut mapping (identity function). For the down-stream tasks, the model uses an approach termed Action Adapter to map the action embedding to the actual actions.

Empirical experiments outperformed the SOTA methods such as EfficientZero (without pre-training) and SGI (pre-training with actions tagged videos) for the custom Atari 50k dataset. The proposed method was pretrained with 1M transition of replay frames from EfficientZero model.

Further experiment was done to pre-train a single model for 6 different games to demonstrate that the model can quickly adapt to different rules with a single pre-trained model. In this setting, the "unified" model outperformed EfficientZero (without pre-training) for 5 out of 6 games.

Ablation studies were also done to demonstrate the importance of the various sub-components (differences reconstruction, vector quantization).

**Summary Of The Review:**

1. Good paper with significant scientific and empirical contributions.
2. However, some experimental setups appear to be incomplete.
3. Overall, the work is of interests to other researchers in the RL and should be accepted for ICLR.

---

### Official Review · Reviewer_HyxY · 2022-10-30

**Confidence:** 4
**Correctness:** 3
**Technical Novelty And Significance:** 3
**Empirical Novelty And Significance:** 3
**Recommendation:** 6

**Clarity, Quality, Novelty And Reproducibility:**

## Clarity
The paper is generally clear and explains the concepts in a well distilled manner. There are certain sections and figures that are unclear and could be fleshed out., like sections 4 and 5, as well as the discussion paragraph. Figures 5 and 6 could be much clearer.

## Quality
The general quality is satisfactory. The analysis are correctly performed, the algorithms are explained and the general concept of the paper appears to be strong. To increase quality, it would be ideal to augment the set of comparisons, flesh out a few missing ablations and improve the final Figures.

## Novelty
The concept introduced appears to be novel. The different moving parts have been seen before, but the combination and subsequent performance improvements appear to be new.

## Reproducibility
The paper is sufficiently clear to be reproducible. Certain details still need to be fleshed out, such as details on loss unrolling and details related to finetuning. More information about how EZ-L was evaluated (and how actions are chosen) would also be ideal.

**Strength And Weaknesses:**

## Strengths
- The paper is clear and understandable
- The propose approach is based on a simple concept, yet appears to generate large benefits when compare to previous techniques.
- The reconstruction losses used in conjunction with FICC seem pertinent.
- The performance increase in the experiments shown is laudable.

## Weaknesses
- Some claims are unclear or not well defined. For example:
     a. unrolling the losses, end of Sec. 3.1 -> how was this implemented?
     b. latent action z has too much information, first paragraph, Sec 3.2 -> can you give information on how you determined this?
- Loss combinations: $L_{cc}$ and $L_{vq}$ appear quite different and could have different scales that can impact training. There is, however, no controlling factors in the final loss. How was this analyzed?
- Writing: some sections could be improved in terms of clarity and grammar, e.g. 4.2, 4.3
- Comparisons: the technique is mainly compared to EfficientZero, but more comparisons to other state of the art techniques would be pertinent.
- Section 4.3 is unclear: are we still building an action adapter M? If not, how do you adapt to the different amount of actions in each environment?
- Ablation studies: there is no ablation showing the individual performance delta of each reconstruction loss (difference reconstruction and full reconstruction) - we only see an ablation related to difference reconstruction
- shortcut learning ablations: there is no analysis showing how reducing the dimensionality of $z_e$ impacts performance. It would be nice to motivate the use of vector quantization instead of just simple dimensionality reduction with quantitative experiments.
- Figure 6 is compelling, but could be much clearer. Labeling each column and highlighting the displacement in the breakout bar would improve its clarity.


## Questions
- Is the Vector quantization enough to stop FICC from converging to trivial solutions? What is the dimensionality of $z_e$ and $z_q$? There could be a world where these vectors, if large enough, still encode enough information to trivially reconstruct $s_{t+1}$, particularly when finetuned on downstream tasks
- Are the action embeddings $e_k$ trained during finetuning? It's clear that R and D are finetuned with learning rate $l_f$, but what happens with $e_k$ is unclear

**Summary Of The Paper:**

The authors propose a pre-training mechanism that improves sample efficiency for model-based reinforcement learning. They propose to use raw videos by extracting action embeddings to pretrain their representations and dynamics. The pretrained networks are then finetuned with limited task data.

In summary, the authors study how pre-training from raw videos can improve model-based RL, propose a method based on cycle-consistency to empower pre-training, and achieve SoTA performance on benchmarks related to Atari games. Emphasis is made on sample efficiency and large improvements over EfficientZero.

**Summary Of The Review:**

The paper proposes a novel idea that shows performance improvements and efficiency gains. In general, the ideas are clearly transmitted and the concept seems simple but powerful. The paper does lack more in-depth comparisons to state of the art methods, and could be clearer in certain sections, but the general idea is sound and interesting. As it stands, the paper is above the acceptance threshold, and could be improved if certain sections are re-written and figures replaced.

---

### Decision · Program_Chairs · 2023-01-20

**Decision:**

Accept: poster

**Justification For Why Not Higher Score:**

Although the reviewers agree that the direction of this paper is exciting and results are convincing, they mentioned that the significance of technical novelty isn't terribly high. Given this, we believe a poster presentation would be appropriate.

**Justification For Why Not Lower Score:**

Unsupervised pretraining for RL is an important area of investigation and this paper proposes a reasonable approach demonstrated by convincing empirical evidence.

**Metareview: Summary, Strengths And Weaknesses:**

The reviews were overall favorable for acceptance. The reviewers found that the paper tackles an important problem that is of wide interest to the community, the proposed technique is reasonable, experiments are extensive and results are convincing, and the paper is generally well-written.

The major strength of the paper is to show that unsupervised pretraining on videos with no state-action labels could improve model-based RL. All reviewers agreed that this aspect is novel and interesting, and that the claims are well justified with convincing empirical evidence.

The reviewers raised some concerns over the technical novelty of the proposed pretraining technique, which is based on well-explored ideas such as reconstruction and cycle-consistency. However, they agreed that their application in the RL setting is interesting. There were also some concerns about the lack of clarity in some of the technical details and results discussion.

The authors' rebuttal has addressed most of the concerns raised by the reviewers. As a result, three reviewers expressed explicit endorsement for acceptance (unfortunately one remaining reviewer wasn't responsive during the discussion period, but upon carefully checking the rebuttal the reviewer's concerns have been addressed adequately).

Given this, we are happy to recommend acceptance.

**Note From Pc:**

if the above contains the word "oral" or "spotlight" please see: "oral" presentation means -> notable-top-5% and "spotlight" means -> notable-top-25%. As stated in our emails, we are disassociating presentation type from AC recommendations